# OpenReview forum: "A Synthetic Dataset for Personal Attribute Inference"
_NeurIPS.cc/2024/Datasets_and_Benchmarks_Track — NeurIPS 2024 Track Datasets and Benchmarks Poster_

### Official Review · Reviewer_RgP5 · 2024-07-02
**Review for "A Synthetic Dataset for Personal Attribute Inference"**

**Rating:** 7
**Confidence:** 3
**Correctness:** Is correct.
**Clarity:** This paper is well written.

**Review:**

Pros:
1. The use of LLM agents to generate synthetic data mitigates privacy concerns and provides a novel approach to dataset creation for PAI research.
2. The synthetic comments are almost indistinguishable from real ones, as demonstrated by human studies, ensuring the dataset's utility.
3. Extensive validation through comparisons with real-world data and across multiple state-of-the-art LLMs reinforces the dataset's reliability.
4. The public release of the dataset and framework supports further research and development in PAI and privacy-preserving methodologies.

Cons:
1. While the dataset includes a diverse range of attributes, it could be expanded to include more complex or nuanced personal information.
2. The quality and diversity of the synthetic data are heavily dependent on the capabilities of the LLMs used, which may introduce inherent biases.
3. Despite the automation, the need for manual verification and labeling of the generated comments is labor-intensive and may not scale efficiently.

**Strengths:**

1. By using synthetic data, the authors effectively address privacy concerns that hinder research using real personal data.
2. The dataset includes a wide range of topics and demographic attributes, making it representative of real-world scenarios.
3. The comprehensive validation through human studies and replication of prior experiments ensures the dataset's credibility and applicability.
4. The framework allows for the scalable generation of synthetic data, which can be adapted for various research needs and extended with additional attributes.

**Additional Feedback:**

N/A

**Documentation:**

The dataset documentation is clear.

**Ethics:**

No ethics concerns.

**Limitations:**

The authors adequately address the limitations and future works.

**Opportunities For Improvement:**

1. It’s better to apply the framework on more social media platform. Future work could include a broader and more nuanced set of personal attributes to enhance the dataset's applicability.
2. Improving the accuracy of automated labeling could reduce the manual effort required and increase scalability.
3. Addressing and mitigating biases inherent in the LLMs used to generate the synthetic data could further improve the dataset's quality and fairness.
4. Incorporating features such as upvotes or comment metadata from real-world platforms could enhance the realism and utility of the synthetic dataset.

**Relation To Prior Work:**

This paper discusses the relation with prior work.

**Summary And Contributions:**

This paper constructs a simulation framework for the popular social media platform Reddit using LLM agents seeded with synthetic personal profiles and generates SynthPAI dataset.
Contributions:
1. A personalized LLM agents-based comment thread simulation framework for producing high-fidelity, diverse, and PAI research-aiding synthetic comments.
2. A curated and hand-labeled synthetic PAI dataset, SynthPAI, of over 7800 comments, creating the first open and privacy-preserving dataset for PAI research.
3. A public release of there framework and dataset SynthPAI and an extensive experimental evaluation showing that SynthPAI is diverse, realistic, and enables insightful PAI research.

---

> ### Author Rebuttal · Authors · 2024-08-16
>
> We would like to thank the reviewer for their time and effort in reviewing our paper, their insightful and constructive comments, and their positive assessment of our work. Below, we address the reviewer's questions and comments.
>
> **Q1: Could the dataset be expanded to more attributes?**
>
> Certainly, and our framework is built to accommodate this. For SynthPAI, we restricted ourselves to these attributes to (1) be comparable to prior work [1] and (2) as the selected 8 attributes already constitute the widest range of attributes in PAI datasets. Notably, for the human verification part of SynthPAI, restricting the number of attributes also helped keep the effort within reasonable (albeit high) levels. While we are very happy with the current dataset and its attributes, we share the reviewer's excitement for more attributes in future work.
>
> **Q2: To what extent do the limitations of the LLMs influence our pipeline?**
>
> We agree with the reviewer that such limitations (i.e., bias, hallucinations) could negatively impact the dataset creation pipeline. While we qualitatively observed these effects to be very little, the human curation steps still helped ensure higher overall quality. Notably, by splitting profile and comment creation, we could, in the first step, ensure that demographic attributes show only little bias and reasonably follow real-world distributions. Naturally, such verification is much harder on free text, where we primarily aim to ensure that comments are consistent with the underlying profile and attribute labels are aligned with human understanding. The reviewer rightfully points out that this human step partially limits the scalability of the approach, however, we remain very hopeful in this regard as (1) not only do models already achieve very high accuracy and have practically saved us significant amounts of time in the creation of SynthPAI but (2) given current scaling trends we expect models to perform even better in future thus reducing the required amount of human effort.
>
> **Q3: Would it be possible in future work to adapt the framework to more social media platforms?**
>
> Absolutely, and we consider this a very interesting item for future work in this area. As of now, we have focussed on Reddit-style threads primarily because they (in the real world) provide highly varied texts in both content and style and are what we consider representative of a wide range of existing multi-person online discussions (especially for PAI research). Nevertheless, the multi-agent approach taken in our framework is conceptually adaptable to other social media platforms where it can be equally used to produce high-quality synthetic discussions.
>
> **Q4: Would it be possible in future work to incorporate meta features into the dataset, such as, for example, upvotes?**
>
> Certainly, and this is especially relevant if we want to get even closer to a full simulation of such online platforms. With the scope of our framework being primarily the creation of texts for PAI analysis, we tried to reduce such meta-behavior in order to keep our data creation pipeline straightforward. For a broader study of, e.g., upvoting behaviors or self-references, such meta-features would be promising additions to our pipeline and we believe our framework is extensible in this direction.
>
> **References**
>
> [1] R Staab et al. Beyond Memorization: Violating Privacy Via Inference with Large Language Models. ICLR 2024.

---

### Official Review · Reviewer_TQdq · 2024-07-21
**A Synthetic Dataset for Personal Attribute Inference**

**Rating:** 7
**Confidence:** 4

**Review:**

This submission addresses a critical privacy concern in the field of Large Language Models (LLMs): their ability to infer personal information from online texts. The authors correctly identify the challenge of creating datasets containing Personally Attributable Information (PAI) when using real data can itself cause harm, effectively demonstrating the value of using simulated data. The main contributions of this work are twofold: (1) an approach for simulating data with synthetic personal profiles attached, and (2) a dataset of 7,800 comments constructed using this approach.

The manuscript is well-written, with a clearly delivered description of the methods and results. The authors do an excellent job referencing similar work and describing the state of the field, including the difficulties faced by research of this kind and the shortcomings of existing approaches and datasets. The appendices are also comprehensive.

The structure of the simulated data is compelling, closely resembling the format of Reddit data (notwithstanding the limits identified by the authors. The attention to diversity in the dataset is particularly commendable.

However, there are some concerns with the approach for constructing the synthetic profiles. While the researchers start from a combination of 8 target attributes, ensuring diversity across these attributes, the use of GPT-4 to fill out the profile and writing style risks perpetuating stereotypes and biases embedded in the model. It would be beneficial to know what steps, if any, have been taken to identify and/or avoid these biases. One other concern with the data generation: the income level is US-specific whereas the place of birth is international, leading to likely unrealistic correlations between those two attributes.

The authors' claim that "average humans can barely outperform random guessing when tasked with distinguishing our comments from real ones" (line 54) is interesting, but it raises the question of whether this is also the case with real data. Some evidence to support this comparison would strengthen the argument at that point in the paper.

Regarding the "fitness for PAI" requirement (R4, line 135), the intuition is correct. However, the question remains: can the researchers confirm that the relationship between text snippets and PAI is the same between the simulated data and real data? In other words, are the PAI clues embedded in the simulated text snippets the same kind as the clues embedded in real text? This aspect is crucial for the validity of the dataset. While the authors include profile and writing style information in the LLM prompt to satisfy R4, this process is black-boxed. I’m just concerned that there’s too much reliance on prompt engineering here. There is no assurance that the simulated comments reflect corresponding PAI in the same manner as real comments, or indeed, that they reflect the corresponding PAI at all. The experimental results of section 4.2 are promising but do not fully confirm this, as a similar difficulty level does not guarantee similar underlying relations in the data.

The thread simulation process could be clarified, particularly regarding how agents determine interest in participating in a given thread (line 179).

The assessment of diversity of profiles and topics is good. However, when assessing comment and thread quality, a more convincing case could be made by comparing simulated comment dataset statistics with statistics of real Reddit data (e.g., how does the distribution of comment length compare between simulated and real?). While the results of the human study differentiating SynthPAI from real data (51.9%) are compelling, it would be interesting to see the results of an LLM study. Can today's LLMs differentiate the posts? This is particularly relevant as LLMs are the ones being evaluated, and the motivating reason for doing this evaluation is that LLMs may be better than humans at identifying PAI.

The limitations identified in the study are reasonable and signal promising future directions of research.

Despite some of the potential issues identified here, this is a strong study with valuable contributions to the fields of evaluation and privacy. The authors have successfully addressed a significant challenge in creating datasets for PAI research and have provided a promising approach and dataset for future work in this area.

**Strengths:**

- Provides a comprehensive and well-structured description of their methods and results.
- Demonstrates careful attention to diversity in the dataset, ensuring a wide range of synthetic profiles and topics.
- Much attention to assessing the dataset's quality, including assessments of diversity, realism, and fitness for PAI inference tasks. This includes using human studies.
- Offers a valuable tool for studying PAI inference without compromising real individuals' privacy.

**Additional Feedback:**

None. thanks for the chance to read this work

**Clarity:**

The submission is very well written with clear indication of the background, motivation, contributions, and results. It is also comprehensive.

**Correctness:**

I found no issues with the dataset generation approach nor with the assessment of the produced data

**Documentation:**

The dataset and code are publicly available online, as shown in the submission. The appendices are comprehensive in their description of the method and results. The inclusion of a datasheet is also appreciated, and it is comprehensive as well.

**Ethics:**

There are no major ethical issues with the research itself. The human study was conducted with IRB clearance. The only concern is the potential for bias and stereotypes in the production of profiles and corresponding comments.

**Limitations:**

The limitations as described by the authors are reasonable. The only limitation I find missing is that this technique relies heavily on prompt engineering, which is not fully verifiable.

**Opportunities For Improvement:**

- Clarify the process of how agents determine interest in participating in a given thread during the simulation (line 179).
- Provide a more detailed explanation of how the profile and writing style information is incorporated into the LLM prompt to satisfy the "fitness for PAI" requirement (R4).
- Address potential biases and stereotypes that might be perpetuated through the use of GPT-4 in constructing synthetic profiles and writing styles
- Include a comparison of simulated comment dataset statistics with statistics of real Reddit data to further validate the quality of the synthetic data.
- Conduct an LLM study to complement the human study in differentiating SynthPAI from real data, as LLMs are the ultimate target of this evaluation.
- Provide more evidence that the relationship between text snippets and PAI is similar in both simulated and real data.

**Relation To Prior Work:**

The paper is strong in this area, with good discussion of previous work and relation to other studies, particularly Staab et al

**Summary And Contributions:**

This submission introduces a novel approach to creating synthetic datasets for studying Personally Attributable Information (PAI) inference in Large Language Models (LLMs). The authors present SynthPAI, a dataset of 7,800 simulated comments designed to mimic real online discussions while preserving privacy. This work addresses a critical challenge in AI privacy research: the need for realistic datasets that don't compromise individuals' personal information.
The authors' main contributions are twofold. First, they propose an innovative methodology for generating synthetic data with attached personal profiles, allowing for the creation of privacy-preserving yet realistic datasets. Second, they provide the SynthPAI dataset itself, demonstrating the practical application of their approach.

---

> ### Author Rebuttal · Authors · 2024-08-16
>
> We thank the reviewer for the effort they put into providing a review with insightful and constructive feedback, as well as their favorable assessment of our work. Below, we address their questions and comments.
>
> **Q1: What measures have you taken to encourage the accuracy, representativeness, and diversity of the generated synthetic profiles?**
>
> We agree with the reviewer that this is a key question when generating synthetic profiles. Notably, while we already saw highly diverse comments across a wide range of demographics using our profile generation step (Appendix D.1), we additionally included a human post-processing step for each profile to ensure that they are sensible, diverse, and aligned with human judgment. Ultimately, the overall dataset demographics (presented in Appendix A.3 and Fig. 9/10) highlight that this process resulted in a diverse and realistic set of synthetic profiles.
>
> As pointed out by the reviewer, one interesting observation we made during post-processing is that models sometimes struggle to correctly derive income levels for specific occupations in particular regions of the world. We remedied this by directly consulting the real-world demographics and adapting the data correspondingly.
>
> For the comment level labels generated via LLM inference, we (despite the high baseline accuracy of LLM inferences) manually verified every single comment label in SynthPAI to ensure their alignment with human intuition. While we do not believe this to be conceptually necessary, we wanted to ensure that the released version of SynthPAI included human-supervised labels. Nevertheless, we believe that with the rapid progress in LLM capabilities, the need for such supervision will decrease noticeably, reducing the need for human effort in future applications of our framework.
>
> **Q2: Could you provide more details on the outcome of the human experiment? Can you repeat the experiment using LLMs as judges?**
>
> We are happy to provide the reviewer with more detailed information about our user study. We note that, in the context of our work, we were primarily interested in average accuracy and FPR (how often humans mistook synthetic texts for real ones). We believe that conducting the same experiment using only human-written texts would be risky from an experimental design perspective as we would falsely trick humans into thinking that synthetic texts are present. Below, we provide more metrics from the original human study (supplementary to Table 2 in the paper):
>
> | Model name | Accuracy | FPR | FNR |
> |:--|:--:|:--:|:--:|
> | Human | 51.9% | 79.2% | 17% |
>
> Per the reviewer's suggestion, we have also conducted the same experiment with LLMs: GPT-4 (the same checkpoint used for generating the dataset) and Llama3-70B. We observed similar results as in the original study with humans (around 50% accuracy, high FPR):
>
> | Model name | Accuracy | FPR | FNR |
> |:--|:--:|:--:|:--:|
> | GPT-4 | 53.3% | 71% | 22.4% |
> | Llama3-70B | 53.4% | 67.4% | 25.5% |
>
> **Q3: Are the types of privacy inferences on the real and the synthetic dataset similar?**
>
> Correct. In addition to the quantitative similarities presented in the paper, we manually verified this by reviewing LLM inferences for real and synthetic data, noticing strong qualitative similarities between them. Exemplary, we below provide the GPT-4 sections of inferences for a synthetic and real comment where the author currently resides in the UK (not disclosing the city for privacy):
>
> *SynthPAI*
> "The author uses the phrase "Bit mad innit?" which is a colloquialism commonly used in the UK. The mention of cyclists and pedestrians also suggests a city environment where such modes of transport are common."
>
> *PersonalReddit*
> "The author of these comments is likely from or currently living in the United Kingdom, based on the use of British English and the mention of a "lift" (which is a British term for an elevator)."
>
> We will highlight more such similarities in the next revision.
>
> **Q4: Can you please clarify how the agents decide if they are interested in engaging in a topic?**
>
> Certainly. After creating the thread topic, we prompt all agents to individually check if they would be interested in engaging with the corresponding thread, given their particular background. We provide the corresponding prompt in Appendix D.2 (Prompt 2) and will make this part clearer in the next revision.
>
> **Q5: Can you please provide a more detailed explanation of how the profile and writing style information is incorporated into the produced comments?**
>
> Of course. Notably, we make sure of two things: (1) Both profile and writing style are always in context when generating a new comment and (2) We created a Chain-of-Thought prompt that requires the model to incorporate (and be consistent) with this information. We present the overall prompt in Appendix D.3 while providing 2 concrete examples of how this is practically instantiated (including model output) in the PDF alongside our rebuttal.
>
> **Q6: How representative is SynthPAI of PersonalReddit on a meta-statistics level?**
>
> This is a very interesting question. Notably, we focused SynthPAI and our pipeline on creating texts fit for PAI research - a task where it performs well in comparison to real-world data [1]. A tradeoff we took to streamline this is that we did not focus on some meta-behaviors appearing on online platforms (upvotes, self-referential behaviors, spam, influence by online trends). On a qualitative level, we also observed a lower rate of highly vulgar/offensive content in SynthPAI than on some subreddits. While in terms of overall demographics and textual statistics (as highlighted by our human study), SynthPAI comments are close to real-world texts such as PersonalReddit [1], achieving meta-statistic similarity remains an exciting area for future research and improvement.
>
> **References**
>
> [1] R Staab et al. Beyond Memorization: Violating Privacy Via Inference with Large Language Models. ICLR 2024.

---

### Official Review · Reviewer_K8Kg · 2024-07-22
**Well written paper with an impactful dataset for PAI research**

**Rating:** 8
**Confidence:** 5
**Correctness:** The claims appear to be correct and r…
**Clarity:** The paper is very well written.

**Review:**

Strengths:

1. The paper is well written and concise. The authors explain the motivation, related work, approach, and evaluation clearly. I enjoyed reading the paper.
1. The proposed dataset is well motivated by the PAI data gap. The ethical and legal concerns enhances the impact of the work.
1. The evaluation of the dataset is extensive and covers most of the facets of the dataset. The IRB-approved user study is also well executed.

Limitations:

1. The use of LLMs for synthetic data generation appears counterintuitive to solve the problem. SOTA LLMs have issues pertaining to hallucination, memorization, PAI, and more. It is unclear how these issues are tackled in the simulation framework.
1. In line with above, the verifiability of seeded LLM agents is missing. All the results are discussed at an aggregate level (e.g., diversity, comment quality). However, how can we ensure that the individual agents are indeed constructing comments according to their synthetic profiles?

**Strengths:**

The submission is a high-quality work with potential of significant impact to the research community. The problem is well motivated and the paper is written in a clear and concise manner. The proposed dataset solves ethical and privacy concerns in the field and enables further research in personal attribute inference.

**Additional Feedback:**

N/A.

**Documentation:**

The paper is very well documented. I would recommend the authors to expand on the ethical and responsible use of the dataset. Additionally, please include a plan for hosting, licensing, and maintenance of the dataset.

**Ethics:**

I don't suspect any ethical concerns in the paper.

**Limitations:**

The authors have adequately addressed the limitations of their work.

**Opportunities For Improvement:**

Please address the limitations mentioned in the review above. Other detailed comments for improvement are as follows:

1. It may be useful to provide an explanation of what personal attribute inference means in the introduction. It is unclear how harmful personal attribute inference can be unless the inference can be used to track down a real person. Additionally, PAI research is mentioned a few times but it is unclear what that entails. Some examples on the applicability of the dataset to real problems may be useful.
1. On a first read, it was a bit unclear that the LLM agents used in dataset generation were different from the LLMs used for generating personal attribute labels. Perhaps a clear distinction in the introduction may help.
1. Why are there only 4748 personal attribute labels for 7823 comments? Is that due to the lack of deducibility of labels for the remaining comments? Additionally, why do the labels go from 4748 to 4730 after human verification?
1. Obtaining SynthPAI requires adjusting model labels. What does that process entail and how many times is an adjustment required in general?
1. Since LLMs are highly agreeable, how is disagreement embedded in the prompt to emulate natural scenarios?
1. The 10% drop in the main experiment in Section 4.2 is fairly substantial. Sharing intuitive reasons for the drop may be helpful.

**Relation To Prior Work:**

Relation to and differentiation from prior work is clearly discussed.

**Summary And Contributions:**

The paper introduces a synthetic dataset, SynthPAI, to advance research in personal attribute inference (PAI). This is motivated by recent studies that have successfully used LLMs to infer personal attributes from online texts. The dataset is generated using a comment-thread based simulation framework that emulates forums like Reddit by seeding LLM agents with synthetic profiles and style anchors.  The synthetic data generated by the framework is further processed by human verifiers resulting in a dataset of 7823 comments. The evaluation of SynthPAI confirms its high fidelity along with alignment of prior PAI results on real datasets.

---

> ### Author Rebuttal · Authors · 2024-08-16
>
> We sincerely thank the reviewer for their time and effort spent reviewing our paper, their insightful and constructive comments, and their overall favorable assessment. Regarding the reviewer's writing and presentation improvement suggestions, we highly appreciate their feedback and will improve the corresponding paragraphs in the next revision of our paper. We address the reviewers' remaining questions and comments below.
>
> **Q1: What impact do the current limitations of LLMs have on the proposed framework and dataset?**
>
> We agree with the reviewer that LLMs have certain inherent limitations. However, in particular for their usage within our pipeline, we have observed little negative impact on them. We attribute this to a few key design choices: (1) We split profile and text generation, including a human verification, ensuring a sound set of initial profiles. (2) We always generate texts with the full grounding of the profile, i.e., the model has the profile in context and is required in CoT fashion to reason about it, closely aligning it to its attributes and style. (3) A final human verification of all resulting texts and labels.
>
> While we took the last step to ensure SynthPAI had high-quality data, we only had to adapt a small fraction of data points manually. We believe the required human effort will strictly decrease with newer text generation models.
>
> **Q2: Can you elaborate on how the agents follow their profiles, style instructions, and criticality grade?**
>
> Certainly. We closely align agents with their characteristics (attributes, style, criticality) by incorporating all three aspects into every comment generation prompt. Furthermore, we require the agents to first reason about their behavior (based on their profile, style, and criticality level) before generating a comment. We provide 2 instantiated examples of this in the PDF (general answer) submitted alongside the rebuttal (which we will include in the next revision of the paper). Empirically, we observe that this strongly aligns the generated texts with the agent profile's underlying characteristics as well as provides human interpretable reasoning chains, which can be used to remedy rare cases with weak profile adherence.
>
> **Q3: How do you end up with 4748 personal attribute labels for 7823 comments? Is that due to the lack of label deducibility for the remaining comments? Additionally, why do the labels go from 4748 to 4730 after human verification?**
>
> Indeed, the reviewer's understanding is correct. As in real world data, a large fraction of comments do not leak personal information, while some can contain more than 1 attribute label. We present a more detailed overview over the number of attribute labels for *individual* comments in the table below:
>
> | \#labels | 0 | 1 | 2 | 3 | 4+ |
> |:--|:--:|:--:|:--:|:--:|:--:|
> | \#comments | 4364 | 2417 | 826 | 203 | 13 |
>
> With respect to the slight decrease after human verification of labels, we find that this is due to the few cases where the model and human do not agree on whether an attribute is inferable (and, as such, some labels get removed during human verification). As automatic tagging of the comments improves with improved versions of LLMs, we believe this discrepancy will decrease. In the next question, we will give more details on the human-in-the-loop adjustment of the automatically obtained labels (Q4).
>
> **Q4: Obtaining SynthPAI requires adjusting model labels. What does that process entail and how many times is an adjustment required in general?**
>
> We will first present the general procedure: A human labeler went through each generated comment by hand (aided by a user interface included in the codebase of the project), which contained fields for all eight possible attributes prefilled with the model inferred values. The human labeler could then adjust each attribute inference individually (including their addition/removal) based on their judgment of the comment. By having a wide range of comments already correctly inferred by the inference model, the overall process was considerably faster per label than obtaining the label from scratch, as in our prior study on real-world data [1].
>
> We included an overview of the statistics of this relabeling process in the main paper in Section 4.1 – Attribute labels and in Table 3. Further, we calculated the accuracy of the tagging model against the final human labels, which amounts to 76%. While this is already relatively high, in order to obtain a high quality dataset we still see human tagging, as performed by us, necessary. However, we believe that with newer and better iterations of LLMs (which can be easily integrated into our framework), the automatic tagging quality will increase, eventually reducing the need for human verification.
>
> **Q5: The 10% drop in the main experiment in Section 4.2 is fairly substantial. Sharing intuitive reasons for the drop may be helpful.**
>
> Indeed, there is a certain numerical gap between the results on SynthPAI and PersonalReddit (real-world comments) in [1], which indicates that SynthPAI poses a greater challenge to LLMs for inference. Based on our experience of working with PersonalReddit, we believe that the reason for this is that humans tend to sometimes leak personal information egregiously, making inference easy for the models. At the same time, the agents in SynthPAI were prompted not to trivially reveal the information included in their profiles, which prevented many "easy" information leaks and led to more difficult examples for inference overall. We note, however, that apart from such trivial cases, the more intricate ways of information leakage are qualitatively similar between the synthetic and the real-world comments.
>
> **References**
>
> [1] R Staab et al. Beyond Memorization: Violating Privacy Via Inference with Large Language Models. ICLR 2024.

---

> > ### Comment · Reviewer_K8Kg · 2024-08-19
> > **Response to Rebuttal**
> >
> > Thank you for addressing most of my concerns and questions. I still advocate for the acceptance of the paper.

---

> > > ### Author Response · Authors · 2024-08-23
> > >
> > > We thank the reviewer for their response acknowledging our rebuttal and are glad to learn that we could address most of their points. In case the reviewer has any outstanding or new questions, we are eager to engage in further discussion.

---

### Official Review · Reviewer_DQhN · 2024-07-23
**SynthPAI enables personal attribute inference**

**Rating:** 7
**Confidence:** 3
**Correctness:** Yes.
**Clarity:** Yes.

**Review:**

This paper explores the ability of LLMs to infer personal information from online texts. Research in this area has been hindered by a lack of suitable personal data. To address this, the authors propose using LLM agents to generate a synthetic dataset for privacy-preserving research. Their evaluation shows that the synthetic dataset is realistic, diverse, and beneficial for PAI research.

**Strengths:**

The motivation of this paper is clear: to provide the first public dataset for PAI research. The paper uses LLM agents to simulate human behavior, generating realistic comments while preserving privacy since no human activity is involved. To enhance the dataset's utility, the authors emphasize that the synthetic dataset should 1) follow the same settings for comments, 2) ensure diversity, 3) maintain quality and style, and 4) be suitable for PAI research. The paper addresses these challenges one by one, allowing readers to follow the reasoning clearly. Extensive experiments show that the synthetic dataset performs similarly to real datasets.

**Additional Feedback:**

N/A

**Documentation:**

Yes.

**Ethics:**

No.

**Limitations:**

Limitations are well disscussed in the paper.

**Opportunities For Improvement:**

The description for Algorithm is quite abstract. Too many notations are introduced in a short paragraph, and it makes readers hard to understand all details without looking for the descriptions multiple times.

I wonder if it's possible to draw a diagram to illustrate the procedure, or maybe providing an example to tell the high level idea.

**Relation To Prior Work:**

Yes.

**Summary And Contributions:**

This paper utilizes LLM agents to generate simulated thread comments, resulting in the creation of SynthPAI, the first open and privacy-preserving dataset for PAI research. Experiments demonstrate that SynthPAI is both realistic and diverse, facilitating insightful PAI research.

---

> ### Author Rebuttal · Authors · 2024-08-14
>
> First of all, we would like to thank the reviewer for their time and effort spent on reviewing our paper and for their overwhelmingly positive assessment of our paper. Additionally, we highly value their constructive comments regarding improving our paper, and will do our best to make the thread generation section clearer in the next revision of our paper. In case the reviewer has any remaining questions, we are happy to engage in a fruitful discussion.

---

### Author Rebuttal · Authors · 2024-08-16

We thank all reviewers for their constructive feedback and their positive assessment of our work. In the rebuttals below, we will address each reviewer's concern individually. Note that alongside this rebuttal, we submit a one-page PDF that contains some examples to aid our clarifications for specific reviewer questions (which we will also include in the next revision of the paper).

---

### Comment · Area_Chair_WcXX · 2024-08-28

Dear Reviewers,

Thank you for taking the time to review this submission. :)

This is a gentle reminder regarding the reviewer-author discussion.

Please respond to the author's rebuttal at your earliest convenience, especially if you have any points of disagreement.

The deadline is August 31 at 11:59 PM AoE!

Early discussion is always appreciated.

Best, AC

---

### Decision · Program_Chairs · 2024-09-26

**Decision:**

Accept (Poster)

**Comment:**

We appreciate the authors' efforts in answering reviewers' questions during the rebuttal phase and to a great extent addressed the reviewers' concerns. The paper received diverse ratings: 7,8,7,7. The reviewers are generally positive about the proposed idea and the problem considered for creating synthetic datasets for studying Personally Attributable Information (PAI) inference in Large Language Models (LLMs). The decision takes into account the paper, the reviews, the rebuttal, and the post-rebuttal reviewer discussion. The AC panel did not find any reason to overturn the majority rating. Based on the above consideration, the AC suggests to accept this paper but the revised version needs to properly include the responses to Reviewers' comments, where possible, the workable links to the dataset and the code.